

# Behavioral flexibility in an invasive bird is independent of other behaviors

Corina J. Logan

Department of Zoology, University of Cambridge, Cambridge, United Kingdom
SAGE Center for the Study of the Mind, University of California, Santa Barbara, CA, United States

## ABSTRACT

Behavioral flexibility is considered important for a species to adapt to environmental change. However, it is unclear how behavioral flexibility works: it relates to problem solving ability and speed in unpredictable ways, which leaves an open question of whether behavioral flexibility varies with differences in other behaviors. If present, such correlations would mask which behavior causes individuals to vary. I investigated whether behavioral flexibility (reversal learning) performances were linked with other behaviors in great-tailed grackles, an invasive bird. I found that behavioral flexibility did not significantly correlate with neophobia, exploration, risk aversion, persistence, or motor diversity. This suggests that great-tailed grackle performance in behavioral flexibility tasks reflects a distinct source of individual variation. Maintaining multiple distinct sources of individual variation, and particularly variation in behavioral flexibility, may be a mechanism for coping with the diversity of novel elements in their environments and facilitate this species' invasion success.

## BACKGROUND

Behavioral flexibility, defined here as changing preferences according to changing circumstances based on learning (*Logan*, *2016a*; *Logan*, *2016b*), is considered a key factor involved in a species' ability to adapt to environmental change (*Lefebvre et al.*, *1997*; *Griffin & Guez*, *2014*; *Buckner*, *2015*; *Chow, Lea & Leaver*, *2016*). However, it is not known how behavioral flexibility works: is it an independent trait, a problem solving ability, does it arise because of links with other behaviors such as neophilia and exploration, or is flexibility the result of an interaction between problem solving ability and other behaviors (see review in *Griffin*, *2016*)? There are a variety of ways to measure behavioral flexibility in an experimental context and all involve allowing an individual to learn about a task, which then changes after the individual becomes proficient. Individuals that adapt their behavior to these changing circumstances are considered to exhibit behavioral flexibility. Paradigms testing behavioral flexibility include tasks such as a multi-access box (*Auersperg et al.*, *2011*; *Manrique, Völter & Call*, *2013*), water tubes (*Logan et al.*, *2014*; *Logan*, *2016a*), and episodic-like memory and future planning experiments (*Clayton & Dickinson*, *1998*; *Dally, Emery & Clayton*, *2006*; *Raby et al.*, *2007*); however, the most widely used measure is reversal learning (e.g., *Bond, Kamil & Balda*, *2007*; *Tebbich, Sterelny & Teschke*, *2010*;

Corresponding author
Corina J. Logan, cl417@cam.ac.uk

*Boogert et al.*, *2011*). Reversal learning involves learning to associate one option with a reward, which subsequently becomes incorrect when the reward is moved to a different option, thus forcing the individual to reverse their preference to consistently obtain the reward. The few studies that have investigated whether behavioral flexibility relates to problem solving ability and speed have found that these traits do not covary in predictable ways (*Boogert et al.*, *2011*; *Griffin et al.*, *2013*; *Isden et al.*, *2013*; *Shaw et al.*, *2015*; *Logan*, *2016a*; *Bebus et al.*, *2016*). Two studies found that faster learners were slower to reverse their preferences (*Griffin et al.*, *2013*; *Bebus et al.*, *2016*), suggesting a speed-accuracy trade off that might depend on an individual's ability to inhibit choosing the previously rewarded response (*Manrique, Völter & Call*, *2013*; *Griffin & Guez*, *2014*; *Liu et al.*, *2016*, but see *Homberg et al.*, *2007*). In contrast, four studies found no correlations between reversal learning speed and problem solving ability or speed (*Boogert et al.*, *2011*; *Isden et al.*, *2013*; *Shaw et al.*, *2015*; *Logan*, *2016a*), which indicates that increased flexibility did not lead to improvements in problem solving. The latter results suggest that flexibility could be a trait that varies across individuals independently of problem solving ability, and all results considered together suggest that variation in flexibility might correlate with other traits that were not measured in these studies.

This leaves an open question of whether behavioral flexibility varies with differences in other behaviors such as exploration, neophobia, risk aversion, persistence, and motor diversity (the number of different motor actions used to attempt to solve a novel problem). There is debate about whether differences in behavior among individuals are linked to suites of correlated behaviors or whether individual behaviors, such as behavioral flexibility, can vary independently (*Coppens, De Boer & Koolhaas*, *2010*; *Cole, Cram & Quinn*, *2011*; *Sih & Del Giudice*, *2012*). Reversal learning is predicted to fall on the fast-slow behavioral type continuum where fast individuals are exploratory, risk seeking, and persistent with poor accuracy because of the speed with which they solve problems, whereas slow individuals are neophobic, risk averse, and more accurately solve problems (*Sih & Del Giudice*, *2012*). Accordingly, slow individuals should be more behaviorally flexible because they might be less impulsive, that is, less likely to rush into a situation and persistently try a particular solution, which gives them more time to survey the environment and attend to the relevant features of the situation (*Sih & Del Giudice*, *2012*). Individuals that are more neophobic and less exploratory would have more time to examine a situation before taking action, thus making them more likely to choose correctly when they do take action (*Sih & Del Giudice*, *2012*).

These predictions are at odds with some evidence from the comparative cognition literature. In a multi-access box paradigm, keas (*Nestor notabilis*) were faster to explore and faster to learn more solutions, as well as faster to switch to trying new solutions when previously rewarded solutions stopped working than New Caledonian crows (*Corvus moneduloides*; *Auersperg et al.*, *2011*). In this case, a positive correlation between flexibility and exploration led to faster problem solving success, rather than the predicted negative correlation that would result from individuals inhibiting their actions and surveying the task. One reason for this difference could be due to it being an interspecies rather than an intraspecies comparison: differences between species are more likely to be larger and easier

to detect than differences between individuals of the same species. Another reason for the disparity between predictions in the comparative cognition and individual differences literatures could come from defining terms differently or not at all. For example, persistence in attempting to solve a task is a measure of impulsivity, however persistence could be defined as (1) the number of attempts directed to all parts of an apparatus or (2) the number of attempts directed at one part of the apparatus before trying a different part of the apparatus. As such, persistence could involve attention to function or not. The second definition might be implied from the individual differences literature, while the first definition might be implied from the comparative cognition literature. It is unclear whether definitional differences might explain opposite predictions because it is only the recent merging of these two fields that has brought about a need to clarify such definitions. Regardless of potential difficulties arising from differences in definitions, if behavioral flexibility correlates with other behaviors, such correlations could mask whether individuals vary in their behavioral flexibility because this trait is independent or because this variation is caused by a correlated behavior (*Herrmann et al.*, *2010*; *Thornton & Lukas*, *2012*; *Seed et al.*, *2012*).

Results from the few studies that investigated the relationship between behavioral flexibility and other behaviors are equivocal. Consistent with predictions, black-capped chickadees (*Poecile atricapillus*) and great tits (*Parus major*) that were more flexible (faster to reverse a previously learned preference) were slower to explore (*Verbeek, Drent & Wiepkema*, *1994*; *Guillette et al.*, *2011*), and great tits that were more flexible (reversal learning) were more neophobic (*Verbeek, Drent & Wiepkema*, *1994*). Two studies provided evidence inconsistent with the predictions that behavioral flexibility will positively correlate with neophobia and negatively with exploration: there were no correlations between behavioral flexibility (reversal learning) and neophobia or exploration in Florida scrub jays (*Aphelocoma coerulescens*; *Bebus et al.*, *2016*), and also no correlations with activity or boldness in wild cavies (*Cavia aperea*; *Guenther et al.*, *2014*). Given this mixed evidence, it is not yet clear whether behavioral flexibility is part of a suite of correlated traits or a trait that varies independently across individuals.

Innovativeness, considered a subcategory of behavioral flexibility, was linked with the number of motor actions used to try to solve a novel problem, but not with persistence or neophobia in several bird species (*Griffin, Diquelou & Perea*, *2014*; *Diquelou, Griffin & Sol*, *2016*; *Griffin & Diquelou*, *2015*). Innovativeness, defined as inventing new behaviors to solve novel problems or using existing behaviors in new ways (*Griffin & Guez*, *2014*), is distinct from behavioral flexibility. For example, great-tailed grackles exhibit behavioral flexibility in two tests involving reversal learning, showing that they are among the fastest bird species to both learn an initial preference and to reverse this preference (*Logan*, *2016a*). However, great-tailed grackles are not particularly inventive when it comes to creating new behaviors to solve novel problems: they did not successfully innovate string pulling or stick tool use, which are behaviors that many other bird species engage in *Logan* (*2016b*). While it is unknown how motor diversity interacts with behavioral flexibility, the prediction is that these traits will positively correlate because increasing the number of motor actions

attempted could increase the probability and speed of finding a successful solution to a novel problem (*Diquelou, Griffin & Sol*, *2016*).

To determine whether behavioral flexibility is related to a variety of behaviors in one species, I investigated great-tailed grackles (*Quiscalus mexicanus*, family Icteridae, hereafter referred to as grackles), a generalist forager (*Skutch*, *1954*; *Johnson & Peer*, *2001*; *Wehtje*, *2003*) that is behaviorally flexible (*Logan*, *2016a*). Grackles are a native invasive species (*Peer*, *2011*): they have expanded their range north from Central America into North America by over 5,500% over the course of 120 years following the expansion of human modified environments, which is their preferred habitat (*Wehtje*, *2003*). Whereas grackles could have expanded their range simply because of an increase in their suitable habitat, species differences in traits that facilitate adapting to environmental change, such as diet, are additionally implicated (*Blackburn, Cassey & Lockwood*, *2009*). Behavioral flexibility is hypothesized to be a mechanism involved in successful species invasions (*Sol & Lefebvre*, *2000*), and a better understanding of how it works could have implications for managing species invasions. To better understand behavioral flexibility, I tested the hypothesis that individual variation in behavioral flexibility correlates with variation in other behaviors on the fast-slow continuum.

I predicted that individuals that were more behaviorally flexible would also be the most neophobic and risk averse, the least persistent and exploratory, and use more motor actions. I quantified grackles' activity levels (exploration) when placed in a novel environment and also measured the amount of time spent in the safest sections of the aviary (risk aversion). I measured grackles' neophobic reactions to a novel object next to a food dish in comparison with controls where only a food dish was present. Persistence and motor diversity were measured from videos of a stick tool use experiment (*Logan*, *2016b*), where no bird successfully invented stick tool use. Therefore, birds were never rewarded for their actions, which is important when measuring persistence because a food reward could differentially influence persistence across individuals: those who are better at the task would receive more food rewards, which might increase their persistence in future trials.

## METHODS

### Ethics

This research was conducted in accordance with the following permits: US Fish and Wildlife Service (scientific collecting permit number MB76700A-0), US Geological Survey Bird Banding Laboratory (federal bird banding permit number 23872), California Department of Fish and Wildlife (scientific collecting permit number SC-12306), and the Institutional Animal Care and Use Committee at the University of California Santa Barbara (IACUC protocol numbers 860 and 860.1).

### Subjects

Eight adult great-tailed grackles (4 females and 4 males) were caught in the wild in Santa Barbara, California and held for 2–3 months in aviaries before being released back to the wild (see *Logan*, *2016a* for full details). Half of the birds were caught in September 2014 and released in December 2014 (Tequila, Margarita, Cerveza, and Michelada; batch 1) and the

other half were caught in January 2015 and released in March 2015 (Refresco, Horchata, Batido, Jugo; batch 2).

### Study set up

Grackles were housed individually in aviaries (183 cm high by 119 cm wide by 236 cm long) at the University of California Santa Barbara. Grackles had *ad libitum* access to water at all times, and unrestricted amounts of food (Mazuri® Small Bird Food) for a minimum of 20 h per day. On testing days, their main diet was removed for up to 4 h while they participated in experiments and could eat bread or peanuts if successful. Apparatuses were placed on tables (60 cm wide by 39 cm long) and rolled into each aviary for sessions (approximately 20 min per session), which were visually isolated from other grackles and video recorded with a Nikon D5100 camera on a tripod. Experimenters stood just outside the aviary door and in full view of the grackles during the persistence and motor diversity sessions, which did not interfere with their behavior (i.e., they readily interacted with the apparatus) because they were habituated to humans in the wild and in the aviary.

### Statistical analyses

Data were analyzed in R 3.2.1 (*R Core Team*, *2015*). For those tests that involved *p*-values, a result was considered statistically significant when $p < 0.05$. When multiple *p*-values were obtained for one experiment, a Bonferroni–Holm correction was applied to avoid obtaining false positive results simply by conducting many tests on the same data.

### Data accessibility

Data are available at the KNB Data Repository (*Logan*, *2016c*; https://knb.ecoinformatics.org/#view/doi:10.5063/F1NS0RSP). Behavioral flexibility data were previously published and are available at KNB (*Logan*, *2016d*).

### Videos

Clips of videos from each experiment are available at: https://youtu.be/aNz7xuECRR0.

### Exploration and risk aversion

The video recorded exploration session lasted 60 min, starting 30 min after a wild bird's release into the aviary, a novel environment. The grackles' previous experience was always the same: they were trapped, blood was collected, and colored rings put on their legs; they were transported to the aviary in a cat carrier in a car, biometrics were taken, and then they were released into the aviary where they were singly housed, given food and water, and the camera was set up outside their door. The camera was restarted every 20 min, otherwise experimenters were out of visual and auditory contact when recording.

Exploration is measured in a number of different ways and I chose two measures for the purposes of this study, which have been used to measure exploration in other species: the amount of activity in a novel environment (exploration; e.g., *Verbeek, Drent & Wiepkema*, *1994*; *Fox et al.*, *2009*) and the amount of time spent in the safe areas of the novel environment (risk aversion; e.g., *Lynn & Brown*, *2009*; *Lerman et al.*, *2012*; *Jolles et al.*, *2014*).

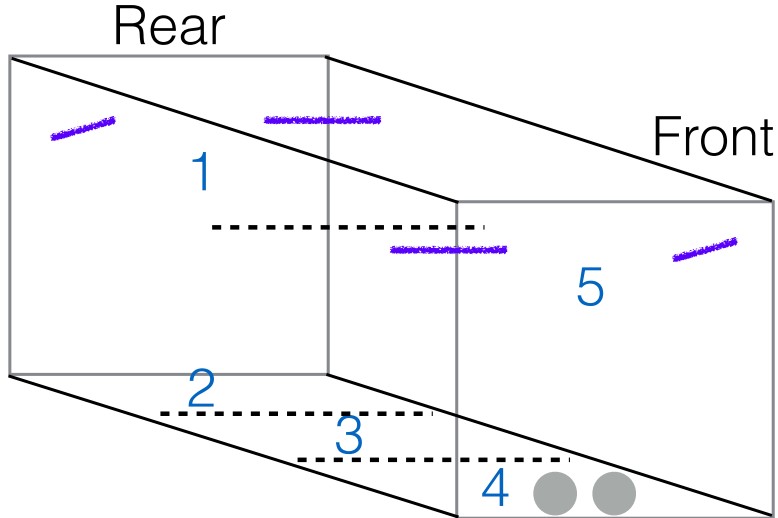

**Figure 1** **Aviary diagram.** The aviary was invisibly sectioned (dotted lines) from least (1–3) to most risky (4–5). The camera was positioned outside a door at the front of the aviary. Food and water bowls were on the ground at the front of the aviary (grey circles) and perches were located in all upper corners (purple lines).

### Exploration

Videos were coded by Linnea Palmstrom to determine how often birds moved between sections of the aviary. The aviary was invisibly sectioned into 5 areas with sections 1 and 5 in the upper half of the aviary, where the perches were located, and sections 2–4 on the ground (Fig. 1). Their exploration score was the number of times they moved from one aviary section to another over the course of the 60 min session, which was an appropriate time period (*Montiglio et al.*, *2010*) because an individual's activity level was ranked similarly regardless of whether activity occurred in the first 5 min or over the whole session (Spearman's rank correlation: $S = 31.43$, $p = 0.10$, rho $= 0.63$, $n = 8$).

### Risk aversion

I (invisibly) divided the aviary into safer versus riskier sections (Fig. 1) and used the percentage of time spent in sections 1–3 (the safer sections) as a measure of risk aversion. The rear of the aviary was considered less risky because it was the farthest from the door where the camera and other equipment were visible, while the ground and the perches in the front of the aviary were more risky because these sections were next to the door and walking on the ground is more dangerous than flying. Food and water were placed on the ground near the door. The aviary was covered in tarpaulins on three sides (both sides and rear), while the side with the door (front) and the ceiling were wire mesh that the bird could see through.

### Neophobia

The neophobia sessions began on a grackle's sixth day in the aviary and involved three 10-min trials with trials 1 and 3 serving as a way to quantify food motivation by placing a food bowl alone on the table, while trial 2 had a novel object 2 cm to the right of the food

bowl (as in *Boogert, Reader & Laland*, *2006*). There were 2 min between trials. In all trials, the food bowl contained 1/4 of a peanut and the latency to land on the table and to feed was recorded as well as which object was approached first in trial 2 (the food bowl or the novel object). Three novel objects were presented in random order to each bird: a GoPro camera inside its clear waterproof case, a stone dropping training apparatus (see *Logan*, *2016a*), and a colored U-tube apparatus (see *Logan et al.*, *2016*). The stone dropping training apparatus was a clear acrylic box (8.8 cm tall by 18 cm wide by 11 cm long) with a clear acrylic tube (9 cm tall, outer diameter = 5 cm) on top. The colored U-tube apparatus was a box (8 cm tall by 40 cm wide by 30 cm long) with a wooden frame covered in cardboard and a clear acrylic top covered by colored paper. Two clear acrylic tubes (both 17 cm tall, one with an outer diameter of 5.1 cm and the other 2.5 cm) protruded from the center of the box and were marked with colored tape at the top. If a grackle did not come to the table within the 10 min period it received a trial duration of 601 s. The neophobia tests were conducted on three consecutive days, with one novel object presented to the bird on each day.

Data were analyzed using the latency to land on the table rather than the latency to feed because birds came to the table more often than they ate the food. The data were not normally distributed (Anderson–Darling normality test: GoPro: $A = 3.08$, $p < 0.001$; stone dropping apparatus: $A = 2.76$, $p < 0.001$; U-tube: $A = 2.46$, $p < 0.001$). Therefore, non-parametric paired Wilcoxon signed rank tests with continuity corrections were conducted to determine whether latencies in control trials (averaged) differed from novel object trials. Neophobia scores were obtained for each novel object and summed for an overall score per individual. Scores were calculated by subtracting the latency to land on the table during the novel object trials (trial 2) from the average latency during control trials (trials 1 and 3). Positive scores indicate less neophobia while negative scores indicate more neophobia. Repeatability of individual neophobic responses across contexts was measured using Spearman's rank correlations to determine whether grackles maintained similar neophobia ranks with each of the three novel objects.

## Persistence and motor diversity

Persistence and motor diversity were calculated as in *Griffin & Diquelou* (*2015*). Persistence was calculated as the attempt rate: the number of times a bird came to the table or interacted with (touched) the apparatus or stick across 21 trials of a stick tool use experiment (105 min/bird; *Logan*, *2016b*). Motor diversity was calculated by counting the number of different motor actions (described in Table 1) performed per individual across the 21 trials of the experiment. Videos were watched from trials 1 to 21 and behaviors from the ethogram (Table 1) were coded at their first observation.

The stick tool use experiment involved an apparatus with a wooden base and rear with clear cast acrylic walls providing a narrow gap at the front and top of the apparatus to insert a stick and retrieve a piece of bread (*Logan*, *2016b*). Birds were given 21 5 min trials to innovate tool use: first, 3 trials with the stick placed on the table next to the apparatus, then 3 trials with the stick inserted into the apparatus, and finally 15 trials with the stick inserted in the apparatus and tool use demonstrated by the human experimenter.

**Table 1  Motor diversity ethogram.** Description of motor actions used while presented with a stick tool use task (techniques 1, 2, 4, 5, 13 and 14 are from *Griffin & Diquelou*, (*2015*) who refer to 'gape' as 'lever').

| Technique | Description | Body part |
|---|---|---|
| 1. Vertical peck | Pecks vertically to the horizontal surface of the apparatus with bill open or closed | Bill |
| 2. Horizontal peck | Pecks horizontally to the vertical edges of the apparatus with bill open or closed | |
| 3. Upside Down Peck | Pecks horizontally to the vertical edges of the apparatus while standing on top of the apparatus, thus the head is upside down | |
| 4. Vertical push | Makes closed bill contact with the horizontal surfaces of the apparatus and slides bill vertically along the surface | |
| 5. Grab apparatus | The apparatus is held between the two mandibles | |
| 6. Grab stick | The stick is held between the two mandibles | |
| 7. Pull stick | The stick is held between the two mandibles and pulled | |
| 8. Push stick | The stick is held between the two mandibles and pushed | |
| 9. Move stick | The stick is moved from inside to outside of the apparatus | |
| 10. Manipulate Stick | Manipulate stick inside apparatus | |
| 11. Carry stick away | The stick is held in the bill as the bird flies away from the table | |
| 12. Throw stick | The stick is tossed to the side | |
| 13. Gape | The closed bill is placed under the edge, in an opening, or on a surface of the apparatus and then opened | |
| 14. Gape upside-down | Same as gape but the head is upside-down (or at least 45 degrees from complete upside-down position | |
| 15. Stand | Stands on top of the apparatus | Feet (or bill) |
| 16. Step | Places one foot on the apparatus | |
| 17. Tips apparatus | Tips apparatus over after standing on top and flying off or by grabbing with bill and pulling over | |

## Measure of behavioral flexibility

These grackles were previously tested on reversal learning of a color discrimination task consisting of a gold tube and a silver tube placed on the table at the same time with one color containing hidden food and the opportunity to make only one choice per trial (*Logan*, *2016a*). Grackles initially learned to search for food hidden in the gold tube and, once proficient, the food was switched to the silver tube and the number of trials required to reach proficiency was assessed. Behavioral flexibility scores were calculated as the number of trials to reverse a color preference minus the number of trials needed to initially learn the color preference. Proficiency in the initial discrimination and reversal was demonstrated if individuals chose correctly in at least 17 of the most recent 20 trials with at least 8 or 9 trials correct per set of 10. I then investigated whether relationships between individual variation in behavioral flexibility and exploration, risk aversion, neophobia, persistence, and motor diversity conformed to predictions.

## General analyses

I determined whether behavioral flexibility (response variable: behavioral flexibility score) negatively correlated with exploration and persistence while examining whether batch had
an effect (explanatory variables) using a Generalized Linear Model (GLM; MCMCglmm function, MCMCglmm package; *Hadfield, 2014*) with a Poisson distribution and log link using 13,000 iterations with a thinning interval of 10 and a burnin of 3,000. The GLM showed acceptable convergence (lag time autocorrelation values were <0.01; *Hadfield, 2010*). Risk aversion and motor diversity were excluded from the analysis because they significantly covaried with exploration and persistence, respectively. A Spearman's rank correlation was used to investigate the relationship between behavioral flexibility and neophobia because residuals were not normally distributed.

Given the small sample size ($n = 7$ for behavioral flexibility scores), I conducted a further analysis to determine whether GLM results were likely to be reliable given the data (*Burnham & Anderson, 2002*). I compared the Akaike weights (range: 0–1, the sum of all model weights equals 1; *Akaike, 1981*) between the test model (above) and a null model (behavioral flexibility score as the response variable and 1 as the explanatory variable) using the dredge function in the MuMIN package (*Bates, Maechler & Bolker, 2011*). If the best fitting model has a high Akaike weight (>0.89; *Burnham & Anderson, 2002*), then it indicates that the results are likely given the data. The null model was strongly supported with an Akaike weight of 0.92, thus indicating the results are reliable even with a small sample size.

### Interobserver reliability

Linnea Palmstrom coded the exploration/risk aversion videos, I coded neophobia videos, and Katherine Lister coded persistence and I coded motor diversity from videos of a tool use experiment (in *Logan, 2016b*). To measure interobserver reliability, I randomly chose 21% of the videos using www.random.org and had a coder who was naïve to the hypotheses (Katharina Brecht) recode their exploration (from which measures of risk aversion are calculated), persistence, and neophobia. I randomly chose three of the eight birds using www.random.org and had Katharina recode their motor diversity (36% of the videos). A higher percentage of motor diversity videos were recoded because agreement determinations were based on the total number of motor actions per bird, which required watching all videos for an individual. There was agreement between Katharina and all other observers for each study: exploration (intraclass correlation coefficient (ICC) = 0.998, 95% confidence intervals (CI) [0.98–1.00]), neophobia (ICC = 0.87, 95% CI [0.67–0.95]), persistence (land on table: ICC = 0.79, 95% CI [0.49–0.93]; interact with apparatus: ICC = 1.00, 95% CI [0.999–1.00]; interact with stick: ICC = 1.00, 95% CI = NA), and motor diversity (ICC = 0.71, 95% CI [0.54–0.82]; ICCs calculated using R package: irr, function: icc, *Gamer et al., 2012*).

## RESULTS

### Exploration and risk aversion

Exploration and risk aversion were significantly negatively correlated, indicating that these two variables might measure opposite ends of the same behavior or an unmeasured behavior might correlate with both and explain their relationship (Spearman's rank correlation: $S = 159.45$, $p = 0.002$, rho $= -0.90$, $n = 8$). To eliminate covariance between

**Table 2  Exploration and risk aversion results.** The percentage of time spent in each aviary section, their risk aversion score (percent time spent in the safest sections of the aviary; sections 1–3) and their exploration score (total number of section changes).

| Bird | Aviary section | | | | | Risk aversion score (% time in safe sections) | Exploration score (section changes) |
|------|------|------|------|------|------|------|------|
| | 1 | 2 | 3 | 4 | 5 | | |
| Tequila | 94 | 0.4 | 0.5 | 6 | 0 | 94 | 16 |
| Margarita | 96 | 0 | 0.1 | 4 | 0 | 96 | 5 |
| Cerveza | 95 | 3 | 0 | 2 | 0 | 98 | 8 |
| Michelada | 92 | 0.06 | 0 | 6 | 2 | 92 | 19 |
| Horchata | 47 | 35 | 5 | 14 | 0 | 86 | 145 |
| Refresco | 100 | 0 | 0 | 0 | 0 | 100 | 0 |
| Batido | 44 | 0.6 | 0 | 0 | 55 | 45 | 30 |
| Jugo | 73 | 12 | 2 | 3 | 11 | 86 | 163 |

explanatory variables, I used exploration to represent this behavior in further analyses (see 'General analyses').

### Exploration

Grackles varied in how many times they changed sections across the 60 min session (0–163), with Refresco having no section changes and Jugo having the most (Table 2). Grackles also varied in the total number of sections they visited during the session (1–5; Table 2).

### Risk aversion

Grackles varied in how much time they spent in the safest sections of the aviary with Batido spending the least amount of time and Refresco the most (Table 2). All grackles (except Refresco) moved through other sections of the aviary and they varied in how much time they spent in sections 4 and 5 (Table 2).

## Neophobia

There were no significant differences between the latency to land on the table in controls (pre [trial 1] or post [trial 3] novel object trials) versus novel object trials (trial 2) (Wilcoxon signed rank tests with Bonferroni–Holm corrected $p$-values: GoPro: *trials 1–2 V* = 21, $p = 1.00$, 95% CI [−283–267], *trials 2–3 V* = 8, $p = 1.00$, 90% CI [−427–277.5]; stone dropping apparatus: *trials 1–2 V* = 7, $p = 1.00$, 80% CI [−369–338], *trials 2–3 V* = 0, $p = 0.54$, 80% CI [−455–(−41)]; U-tube: *trials 1–2 V* = 1, $p = 0.88$, 80% CI [−481–(−85)], *trials 2–3 V* = 1, $p = 0.88$, 80% CI [−507–(−190.5)]). Refresco and Margarita were overall less neophobic than the other grackles, and Horchata was the most neophobic (Table 3). There were many trials in which the bird did not come to the table. However, this did not usually appear to be due to neophobia because it happened in many control trials as well as novel object trials, indicating that it might have been due to a lack of motivation to eat or approach the object.

There was no individual repeatability of neophobia scores across contexts (Spearman's rank correlation with Bonferroni–Holm corrected $p$-values: GoPro vs. stone dropping apparatus: $S = 79.21$, $p = 1.00$, rho $= -0.41$; GoPro vs. U-tube: $S = 56.00$, $p = 1.00$, rho $= 0.00$; U-tube vs. stone dropping apparatus: $S = 20.68$, $p = 0.88$, rho $= 0.63$).

**Table 3 Neophobia results.** Neophobia scores for each novel object and an overall score for each individual. Neophobia score calculations: the latency to land on the table in controls (trials 1 and 3 averaged) minus the latency in the novel object condition (trial 2) for each object type (GoPro camera, stone dropping apparatus, and U-tube apparatus), and summed across object types for the overall neophobia score (positive, less neophobic [bold text]; negative, more neophobic).

| Bird | GoPro | Stone dropping apparatus | U-tube | Neophobia score |
|------|-------|--------------------------|--------|-----------------|
| Tequila | **7** | −444.5 | −156.5 | −594 |
| Margarita | **20** | 0 | 0 | **20** |
| Cerveza | −182 | **167.5** | −42.5 | −57 |
| Michelada | 0 | 0 | −228 | −228 |
| Horchata | −580 | −1 | −277.5 | −858.5 |
| Refresco | **1** | **148.5** | **1** | **150.5** |
| Batido | **187** | −275.5 | −541 | −629.5 |
| Jugo | **338** | −227.5 | −373.5 | −263 |

**Table 4 Persistence and motor diversity results, and behavioral flexibility scores.** Persistence (the total number of times a bird landed on the table, touched the apparatus, or touched the stick), motor diversity (the total number of motor actions used), and behavioral flexibility scores (number of trials to reverse a preference minus the number of trials to initially learn the preference; from *Logan* (*2016a*)) per bird.

| Bird | Sex | Persistence | Motor diversity | Behavioral flexibility score |
|------|-----|-------------|-----------------|------------------------------|
| Tequila | M | 175 | 6 | 70 |
| Margarita | F | 72 | 5 | 70 |
| Cerveza | F | 81 | 2 | 60 |
| Michelada | F | 18 | 1 | 30 |
| Horchata | F | 145 | 8 | 100 |
| Refresco | M | 1,114 | 14 | 50 |
| Batido | M | 4,047 | 15 | – |
| Jugo | M | 197 | 6 | 40 |

**Notes.**
–, did not complete this experiment.

## Persistence and motor diversity

Grackles varied in the number of motor actions they used (1–15) and in how persistent they were (18–4047 = total number of times a bird landed on the table, touched the apparatus, or touched the stick; Table 4). A post-hoc analysis showed that these two variables were significantly positively correlated, indicating that they could have measured the same behavior or have been caused by another, unmeasured variable (Spearman's rank correlation: $S = 8.55$, $p = 0.002$, rho $= 0.90$, $n = 8$). Therefore, only persistence was used in further analyses (see 'General analyses').

## Does behavioral flexibility positively correlate with motor diversity and risk aversion, and negatively with exploration and persistence?

Birds that were more flexible (i.e., faster to reverse a preference: number of trials to reverse a preference minus the number of trials to initially learn the preference) did not have

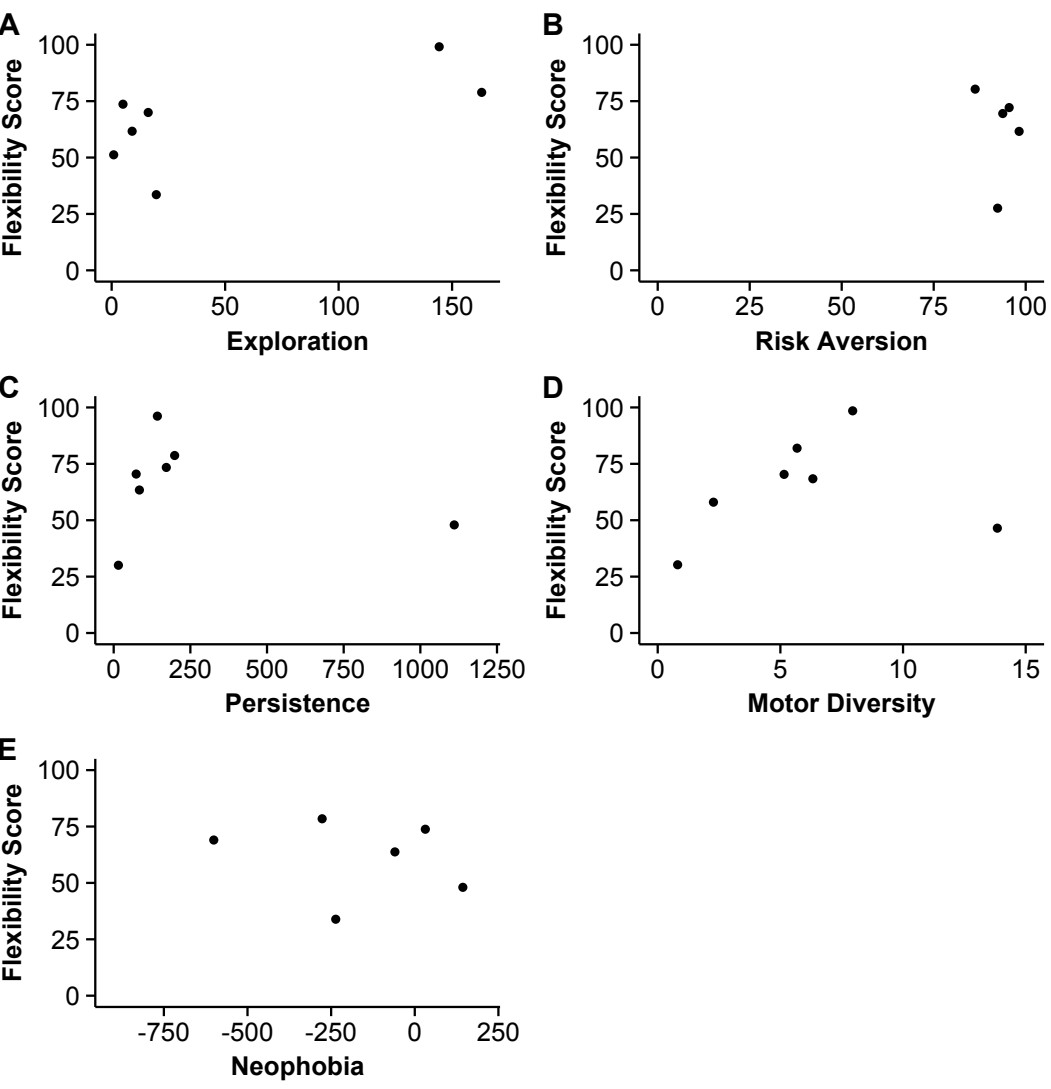

**Figure 2** **Behavioral flexibility scores in relation to other behaviors.** The relationship between behavioral flexibility scores and exploration (A, total number of aviary section changes), risk aversion (B, percentage of time spent in safer aviary sections), persistence (C, total number of interactions with the table, apparatus, and stick), motor diversity (D, total number of different motor actions used), and neophobia (E, latency to land on table during controls minus latency to land next to a novel object) ($n = 7$ grackles).

higher exploration scores, they were not more persistent, and there were no batch effects (Fig. 2, Table S1). There were no significant correlations between flexibility and exploration or persistence, indicating that results did not provide evidence for the predicted negative correlations.

Risk aversion and motor diversity significantly covaried with exploration and persistence, respectively, and these relationships were investigated further. I confirmed that the relationship between these variables and behavioral flexibility was the same as their collinear variables with an additional GLM. This GLM was the same as above, except the

explanatory factors were motor diversity, risk aversion and batch. As above, flexibility did not correlate with risk aversion or motor diversity (Fig. 2, Table S2).

### Does behavioral flexibility positively correlate with neophobia?

Grackles that were more flexible (i.e., faster to reverse a preference) did not have lower neophobia scores, which would indicate more neophobia. There was no significant correlation between behavioral flexibility scores and neophobia (Fig. 2; Spearman's rank correlation: $S = 92$, $p = 0.12$, rho $= -0.65$).

## DISCUSSION

### Exploration and risk aversion

The exploration and risk aversion scores significantly negatively correlated with each other, indicating they might have measured opposite ends of the same behavior. While risk aversion scores could have been confounded by the placement of food and water in a risky section, which might attract birds to this area, they spent only 0–14% of their time in the section with the food and water (section 4). This indicates that they behaved more according to the prediction that this section would be treated as risky even when an attractor was present.

I question whether the measure of exploration actually measured exploration in this species. A bird that is stressed tends to fly back and forth in an aviary, which is not an indicator of exploration, but would be interpreted as such according to the section change measure of exploration. In this study, Jugo mostly flew back and forth between the perches near the top of the aviary while looking up and out of the aviary and not attending to the environment within the aviary. In contrast, Horchata also had many section changes, but she usually walked calmly on the ground, thus perhaps in her case this measure of exploration was appropriate. Therefore, at the species level, activity levels are likely not a good indicator of exploration behavior. Indeed, a distinction is made between forced exploration, where an individual is placed in a novel environment, and voluntary exploration, where an individual in a familiar environment is provided with the opportunity to enter a novel environment (*Guenther, Finkemeier & Trillmich*, *2014*). A study on wild guinea pigs (*Cavia aperea*) found that these two variations of exploration measure different behaviors: forced and voluntary exploration activity did not correlate in juveniles or adults (*Guenther, Finkemeier & Trillmich*, *2014*).

Voluntary exploration would likely be a more accurate measure of actual exploratory behavior in grackles, which could also involve voluntary exploration of a novel object in a familiar environment. Such a measure is also called a neophilia test where a novel object is placed in a familiar aviary in the presence of (but not next to) their regular food source to determine how soon the bird approaches and interacts with the apparatus and for how long (as in *Mettke-Hofmann, Winkler & Leisler*, *2002*). This kind of test would also likely more directly relate to how grackles have expanded their range so rapidly: rather than exploring novel environments, grackles are more likely to have successfully expanded their range by exploring novel objects. Grackles have not necessarily needed to adapt to novel environments during their range expansion because it coincided with an increase in

their suitable (human-managed) habitat (*Wehtje*, *2003*). Exploration is more likely to have played a role in exploiting novel objects in their environment because humans throw away products that may be novel to grackles (e.g., egg cartons, yogurt cups) and design new potential food sources (e.g., dumpsters) where food is not necessarily obvious, therefore the objects must be explored to determine whether they contain food.

## Neophobia

Grackles were not generally neophobic because no significant differences were found between controls and novel object trials in the latency to land on the table. Indeed, the GoPro camera, which was also the smallest of the novel objects, appeared to attract their attention more than the food. Comparing grackles with other species that have been tested using a similar design, it appears that they are less neophobic than starlings (*Sturnus vulgaris*; *Boogert, Reader & Laland*, *2006*), blue tits (*Cyanistes caeruleus*; *Herborn et al.*, *2010*), Japanese quail (*Coturnix japonica*; *Zimmer, Boogert & Spencer*, *2013*), Chimango caracaras (*Milvago chimango*; *Biondi, Bó & Vassalo*, *2010*), European greenfinches (*Carduelis chloris*; *Herborn et al.*, *2011*), Indian mynas (*Sol, Griffin & Bartomeus*, *2012*; *Griffin & Diquelou*, *2015*), and mountain chickadees (*Poecile gambeli*; *Kozlovsky, Branch & Pravosudov*, *2015*), and that they have similar levels of neophobia as noisy miners (*Manorina melanocephala*; *Griffin & Diquelou*, *2015*) and a different group of mountain chickadees (*Fox et al.*, *2009*).

## Persistence and motor diversity

The persistence and motor diversity scores significantly correlated with each other, indicating they might have measured the same behavior or have been caused by another, unmeasured variable. This suggests that the longer a bird persists in attempting to solve a task, the more likely it is to use a wider variety of motor actions. Therefore, it is likely that individuals that used few motor actions would likely have used more if they were perhaps more motivated to interact with the task. Measuring persistence and motor diversity in a variety of contexts could address this potential issue and clarify whether these variables actually do covary on a task that all individuals persist on. These results are different from findings using a similar experimental design on Indian mynas and noisy miners where motor diversity differed between species, but persistence did not, thus indicating these were two separate behaviors (*Griffin & Diquelou*, *2015*). However, birds in *Griffin & Diquelou* (*2015*) could receive food rewards from the apparatus if successful (i.e., at the end of each bout of persistence) and such positive reinforcement for persisting could have increased persistence for successful individuals in subsequent trials.

## Behavioral flexibility

Contrary to predictions, behavioral flexibility did not correlate with exploration, neophobia, risk aversion, persistence, or motor diversity. The small sample size might have limited my ability to detect significant correlations; however, the behavior of the models suggested this was not the case. It is perhaps not surprising that behavioral flexibility did not correlate with neophobia (the only behavior I was able to obtain repeatability measures from) because neophobia was not expressed consistently across contexts. This could indicate a further source of individual variation in grackles or it could result from the inability of the method

to accurately measure neophobia in this species. The latter highlights the importance of conducting repeatability tests when attempting to understand how two variables correlate because two unpredictable variables (behavioral flexibility and neophobia) would not likely correlate with each other. These results are similar to results from Florida scrub jays where behavioral flexibility (reversal learning) did not correlate with neophobia or exploration (*Bebus et al.*, *2016*). The few studies that investigate the relationship between behavioral flexibility and other behaviors either show relationships opposite to predictions (*Verbeek, Drent & Wiepkema*, *1994*; *Guillette et al.*, *2011*) or show no correlations (this study, *Bebus et al.*, *2016*). One prediction was supported in only one test: reversal learning speed negatively correlated with neophobia (*Verbeek, Drent & Wiepkema*, *1994*). This accumulating evidence suggests the need to reconsider the basis for hypotheses linking other behaviors with behavioral flexibility.

## CONCLUSIONS

Traditionally, behavioral flexibility is thought of as a cognitive ability (see review in *Shettleworth*, *2010*) and is considered as such in hypotheses linking it with other behaviors (*Sih & Del Giudice*, *2012*; *Guenther & Trillmich*, *2013*). However, mixed results, with none conforming to predictions, from grackles, keas and New Caledonian crows question this assumption. Grackles lacked correlations between behavioral flexibility and problem solving ability and speed, and individuals that were behaviorally flexible in one type of test were not necessarily flexible in a different type of test (*Logan*, *2016a*). The more exploratory keas were more behaviorally flexible on a multi-access box and faster to innovate new solutions to novel problems than New Caledonian crows (*Auersperg et al.*, *2011*). These mixed results indicate a need to look beyond cognitive and behavioral measures that might correlate with behavioral flexibility and investigate relationships with factors such as physiology and genetics. For example, grackles that are in better phenotypic condition (e.g., have better immunity) might have the capacity to be more behaviorally flexible than individuals in worse phenotypic condition. Non-behavioral, non-cognitive individual factors have yet to be measured in relation to behavioral flexibility. Considering behavioral flexibility in this more integrated way could allow experimenters to manipulate this elusive trait to understand what it is and how it works.

Though the sample size is small, these results provide further support that behavioral flexibility represents a distinct axis of individual variation in behavior. Behaviors that do not correlate with each other are suggested to represent "inherent individual differences" in each of the traits measured (*Cole, Cram & Quinn*, *2011*, p. 495). For example, great tit problem solving ability did not correlate with body condition, neophobia, or exploration; therefore problem solving was considered its own behavior that varies across individuals rather than varying due to links with other individual traits (*Cole, Cram & Quinn*, *2011*). The methods used to measure neophobia and exploration in grackles might not have accurately represented these behaviors, therefore further investigations using different methods that are validated measures of these behaviors in grackles should be explored before entirely ruling out correlations with behavioral flexibility. Previous research on

grackles and other species has shown that behavioral flexibility is independent from innovativeness (*Logan*, *2016b*), problem solving ability and speed (*Boogert et al.*, *2011*; *Isden et al.*, *2013*; *Logan*, *2016a*), or that it negatively correlates with problem solving speed (*Griffin et al.*, *2013*). The majority of evidence so far indicates that individual variation in behavioral flexibility is not confounded with other behaviors, although two alternative hypotheses cannot yet be ruled out: the behaviors might not have been measured with enough consistency across studies to directly compare the results, or the behaviors are not repeatable enough within individuals to reliably covary with each other. Further research is needed to distinguish which hypothesis is supported.

It could be adaptive for invasive species, such as the grackle, to maintain many independent axes of individual variation and, in particular, variation in behavioral flexibility. Indeed, Western bluebirds rely on existing intrapopulation variation when expanding their range (*Duckworth*, *2008*). While Western bluebirds rely on variation in dispersal strategies, grackles may rely on maintaining individual variation in behavioral flexibility, which could allow them to more quickly adapt to changing or unpredictable environments.

## FUTURE DIRECTIONS

Future research investigating neophobia, exploration, persistence and motor diversity in this species would benefit from a larger sample size, replicability of results from multiple groups, and finding measures that are repeatable within individuals to determine the reliability of these conclusions. Incorporating the use of a factor analysis would help determine whether correlated explanatory variables measure distinct behaviors or arise from unmeasured correlated variables. Investigating each variable using multiple methods will facilitate an understanding of which methods actually measure the behaviors of interest.

## ACKNOWLEDGEMENTS

I am grateful to Linnea Palmstrom for coding the exploration videos; Katherine Lister for coding the persistence videos; Katharina Brecht for coding all videos for interobserver reliability analyses; Luisa Bergeron, Christin Palmstrom, Linnea Palmstrom, and Michelle Gertsvolf for trapping and aviary assistance; Steve Rothstein for use of the aviaries; Joe Jablonski for making the cast acrylic apparatuses; Jill Zachary and Kathy Frye at Santa Barbara City Parks and Recreation for use of the Andree Clark Bird Refuge and East Beach Park; Estelle Sandhaus and Chris Briggs at the Santa Barbara Zoo for access to wild grackles; Dieter Lukas, Ted Bergstrom, and Carl Bergstrom for discussions; Alecia Carter for analysis and manuscript feedback; and Jennifer Vonk, Lily Johnson-Ulrich, and Andrea Griffin for comments on a previous draft.

### Funding

Funding was provided by the National Geographic Society/Waitt Grants Program (grant number W252-12) and Junior Research Fellowship, the SAGE Center for the Study of the Mind at the University of California Santa Barbara. The funders had no role in study design, data collection and analysis, decision to publish, or preparation of the manuscript.

### Grant Disclosures

The following grant information was disclosed by the author:
National Geographic Society/Waitt Grants Program: W252-12.
Junior Research Fellowship, SAGE Center for the Study of the Mind, University of California Santa Barbara.

### Competing Interests

The author declares there are no competing interests.

### Author Contributions

- Corina J. Logan conceived and designed the experiments, performed the experiments, analyzed the data, contributed reagents/materials/analysis tools, wrote the paper, prepared figures and/or tables, reviewed drafts of the paper.

### Animal Ethics

The following information was supplied relating to ethical approvals (i.e., approving body and any reference numbers):

This research was carried out in accordance with permits from the US Fish and Wildlife Service (scientific collecting permit number MB76700A), California Department of Fish and Wildlife (scientific collecting permit number SC-12306), US Geological Survey Bird Banding Laboratory (federal bird banding permit number 23872), and the Institutional Animal Care and Use Committee at the University of California Santa Barbara (IACUC protocol number 860 and 860.1).

### Data Availability

Great-tailed grackle neophobia, persistence, exploration, risk aversion, motor diversity, interobserver reliability data and R code, Santa Barbara, CA USA 2014–2015. KNB Data Repository, DOI:10.5063/F1NS0RSP; https://knb.ecoinformatics.org/#view/doi: 10.5063/F1NS0RSP.

### Supplemental Information

Supplemental information for this article can be found online at http://dx.doi.org/10.7717/peerj.2215#supplemental-information.

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
