# Peer review of "Behavioral flexibility in an invasive bird is independent of other behaviors"

_PeerJ, doi:10.7717/peerj.2215_

## Round 0.1 · original submission · Major Revisions

Decision Letter for Logan, “Behavioral flexibility in an invasive bird…”

I have now received two reviews from experts on the topic of your MS. Both see the value in your study and appreciate several aspects of the study, but both also indicate fairly significant concerns about the clarity of your theorizing and constructs. They have fairly specific recommendations for your paper, which I do not need to reiterate here, but I share their emphasis on the importance of taking care to provide your own operational definitions and to be clear on how these may be similar or different from other uses of the terms in the literature. Whereas you cannot also easily increase your sample size, I would also encourage you to be circumspect in the conclusions you draw given your relatively small sample size. I have a few additional comments of my own.

Given your measure of reversal learning, I’m surprised you do not discuss inhibition as a key factor relating to behavioral flexibility. Animals who are quick to show reversal of a previously learned response are likely strong inhibitors, which may also relate to a lack of exploration in a novel circumstance, explaining what you describe as an unexpected finding in lines 35-37. I would think that the link between reversal learning and neophobia cited on lines 41-42 is actually consistent with the previous finding, although you list them as discrepant. Overall, I think the introduction needs more tightening and you could be more specific about how you are testing your hypotheses, as well as a little more background on the grackle.
On lines 93-94 be clear if the controls refers to objects or other birds.
More details are needed on the tool use task because it is not yet available to readers. Even though the color discrimination task results are now available to readers, it would also help to have more details of this task in the current MS making clear what is novel about the current study. In general I would advise against mining the same data for multiple papers.

Ideally, all raters would be naïve to the hypotheses of the experiment. If you are to sample a proportion of the video for reliability, it should be a random sampling (e.g. not just from a single subject) and should probably account for at least 20% of the video (preferably more). It is important that you have another coder (other than yourself) match interobserver reliability for motor diversity. I will request that you supplement the observer reliability data to meet these aims before resubmission. If you need an extension please let me know as I realize this can be time-consuming.

If the food and water was placed on the floor, wouldn’t this encourage birds to be in risky areas in Exp. 1?

It strikes me as a bit odd to submit Exp. 1 as a study of exploration. Given that you immediately discount your own measures as being valid, and given that there is no relating exploration to any other variables, it seems more like a pilot assessment of a measure than an actual Experiment. If you still have access to the subjects of the study, ideally you could conduct a different measure of exploration, or omit altogether given that you have the neophobia data as well. Similarly, I would not consider an assessment of neophobia to be “Exp. 2” as nothing is being manipulated per se. You are simply assessing variation in a trait between subjects for use in a later analyses. These components of the study would be better treated as assessments than as experiments. The entire MS could be written as a single study (with ‘study’ being more appropriate than ‘experiment’ given that behaviors are being assessed and correlated rather than manipulated).

It would be useful to have a future directions section where you outline the ideal approach to this research question indicating the need for a larger N and multiple methods of assessing each factor.
Figure 1 would be more useful to show a grid system clearly indicating sections, which when overlain over the video, could be used by the coder to determine which section the bird was in during an observation interval. The current figure is not very informative. Include locations of feeding dishes, perches etc. in a revised figure. In reading the text, it was difficult to envision how the raters scored transitions.

I don’t think the final figure is necessary.

Please make sure that figure 2 uses the same axes for all panels (e.g. max of 100). Please align the panels exactly.

·

Basic reporting

This submission conforms to PeerJ policies. The author uses clear, unambiguous professional English language. The introduction and background put the article strongly into context and and identify an existing knowledge gap. The author focuses on behavioral flexibility as measured exclusively by reversal learning, but should mention that there is a great deal of literature on behavioral flexibility and its underlying cognitive and non-cognitive processes outside of reversal learning (e.g. Herborn et al., 2014).
The structure does not conform to PeerJ standard and the manuscript layout is somewhat difficult to follow. The author should put methods, results, and discussion into single sections with subheadings for the different experiments in order to improve clarity.
Figures and tables are mostly relevant and high quality. Figures 1, 2, and 3 lack figure legends but are otherwise relevant and high quality. Figure 3 is not mentioned in text and is also lacking a figure legend. Without either I cannot say that Figure 3 is relevant or high quality.
I would suggest the addition of a table or figure for the introduction that outlines the predictions about which behavioral measures are thought to correlate with fast and slow behavioral types and behavioral flexibility (e.g. Figure 1 from Réale et al., 2010 or Table 1 from Sih & Del Giudice, 2012).
Raw data is only supplied for experiment 2 (neophobia). There is an indirect link to the data for behavioral flexibility. There is no link to the raw data for exploration, risk aversion, motor diversity, or persistence. Even though some of this data has been previously published, the author should include direct links to all data used in this study.

Herborn, K. A., Heidinger, B. J., Alexander, L., & Arnold, K. E. (2014). Personality predicts behavioral flexibility in a fluctuating, natural environment. Behavioral Ecology, 25(6), 1374–1379. http://doi.org/10.1093/beheco/aru131
Réale, D., Garant, D., Humphries, M. M., Bergeron, P., Careau, V., & Montiglio, P. O. (2010). Personality and the emergence of the pace-of-life syndrome concept at the population level. Philosophical Transactions of the Royal Society of London. Series B, Biological Sciences, 365(1560), 4051–63. http://doi.org/10.1098/rstb.2010.0208
Sih, A., & Del Giudice, M. (2012). Linking behavioural syndromes and cognition: a behavioural ecology perspective. Philosophical Transactions of the Royal Society of London. Series B, Biological Sciences, 367(1603), 2762–72. http://doi.org/10.1098/rstb.2012.0216

Experimental design

This is original research within the scope of the journal. The research question is well defined; it is not clear whether behavioral flexibility is related to suites of cognitive or non-cognitive traits or neither and it may in fact be an independent dimension. The author overall does a good job of designing an experiment to address this question and this study contributes to filling the introduced knowledge gap. The methods are described with sufficient detail for replication and the authors met prevailing ethical standards and included a statement of ethics and detailed descriptions of the treatment of subjects. I did have some concerns about a few of their behavioral measures, but the author included a highly detailed description of potential design flaws and the steps taken to control for them. The author was open about which measures might not be representative for grackles (e.g. exploration and neophobia), but used very well accepted standards for measuring these behaviors and followed meticulous protocol.
One measure I had concerns about in terms of design was for risk aversion as there was only one citation referencing this method and it referenced a study using sticklebacks (not birds). Sticklebacks are a highly studied model organism and what constitutes a risky behavior for them is well known. However, what might be a risky behavior for a grackle (this study) was not as convincing. A measure of risk aversion that uses a dummy predator is a well-accepted standard with birds and other species. Here, simply including one or two more citations for this method, preferably of quantifying risk aversion in birds, would be satisfactory to address this concern.

Validity of the findings

The authors found no relationship between behavioral flexibility and their five non-cognitive behavioral measures. Overall, their conclusion that behavioral flexibility may represent a distinct axis of variability that is unrelated to behavioral syndromes is valid. However, I would recommend a more conservative approach with regards to generalizing the conclusion that behavioral flexibility is also distinct from other cognitive traits outside of grackles. Here, the author should mention that some studies do find a relationship between behavioral flexibility, problem-solving, and innovation in species other than grackles (e.g. Auersperg et al. , 2011; Deaner et al., 2006; Leal & Powell, 2012).
Though their experimental design for exploration and neophobia was sound, given the concerns the author mentioned in the manuscript I would also caution against drawing strong conclusions from this data. Exploration might be better described as simply “activity” and the author should remind readers in the discussion section that the lack of significant results may be due to the fact that the author suspects this way of measuring exploration may not be a good indicator of exploration for grackles.
Likewise, given that birds did not differ in control or test conditions for examining neophobia, and did not correlate across tests, it might not be valid to use a batch neophobia score to compare to behavioral flexibility. Here, I would also remind readers that the lack of significant results could be due the fact that neophobia scores may not actually reflect real variation in neophobia between grackles. Given these considerations I would suggest being more conservative in the interpretation of these null findings, at least for exploration and neophobia.
Overall, their use of statistics was sound and they gave full consideration of their model fit and the reliability of their results given their small sample size. This work challenges the idea that behavioral flexibility is part of a syndrome of personality and non-cognitive traits and that behavioral syndromes can explain variation in cognitive and non-cognitive behaviors.

Auersperg, A. M. I., von Bayern, A. M. P., Gajdon, G. K., Huber, L., & Kacelnik, A. (2011). Flexibility in problem solving and tool use of kea and New Caledonian crows in a multi access box paradigm. PloS One, 6(6), e20231. http://doi.org/10.1371/journal.pone.0020231
Deaner, R. O., van Schaik, C. P., & Johnson, V. (2006). Do some taxa have better domain-general cognition than others? A meta-analysis of nonhuman primate studies. Evolutionary Psychology, 4, 149–196.
Leal, M., & Powell, B. J. (2012). Behavioural flexibility and problem-solving in a tropical lizard. Biology Letters, 8(1), 28–30. http://doi.org/10.1098/rsbl.2011.0480

·

Basic reporting

My comments on reporting are included in the comments to authors

Experimental design

The design is suitable except that the sample size is small

Validity of the findings

My comments are included in those below

Additional comments

This article reports the results of a series of correlations between reversal learning (referred to as behavioural flexibility in the paper) and collection of personality scores in a small sample of grackles (N=8). The author finds no evidence of any relationships leading her to suggest that behavioural flexiblity is an independent trait. The results contribute to a growing body of work investigating this type of relationship. I found the paper and its results interesting.

My primary concern is that the author has no evidence that inter-individual variation in the target behaviours (e.g. neophobia, exploration etc..) is actually repeatable across time and/or contexts. It is therefore dubious to what extent the lack of correlation is actually due to a lack of correlation or simply to the fact that the tests are not providing a reliable measure of an individual’s behaviour. It should be noted that the sample size falls very substantially short of the sample sizes recommended for this kind of study.

My second concern is that the that there is no rigorous use of the term behavioral flexibility; the text begins by defining the trait as learning, but then moves on to equate it to reversal learning, and then goes on to draw parallels between the present findings and other studies on ‘behavioral flexibility’ which use problem solving tasks (and not reversal learning) to measure behavioral flexibility (e.g. Cole et al. 2011). Hence, the lack of any systematic patterns might be more due to the lack of consistency in definitions and measurements rather than to independent evolution. Overall, the paper suffers from the same weaknesses as many papers in this field, which is loose logic, referencing and descriptions of past work. Whereas I feel my first concern cannot be addressed given the nature of the study, the second could be addressed in a major revision. The paper could do with a thorough read before re-submission to eliminate typos.

Detailed comments are as follows:

L10: Griffin & Guez 2014 never defined behavioral flexibility in this way. On the contrary, they proposed that the trait is multi-dimensional. Please cut the reference to Griffin & Guez 2014. To my knowledge, there is no ‘official’ definition of the concept. Behavioral flexibility is just a term that began being used in association with the large scale comparative literature on innovation anecdotes. I suggest it would be more appropriate to cut the definition or to state clearly that this is the definition proposed by the author. I also refer the author to two recent publications by Sol et al. and Griffin in Phil Trans for alternative views on the nature of behavioral flexibility.
L12: why yet? It is not clear why the proposed definition has for consequence predictable relationships between behavioral flexibility and problem solving. I suggest the logic needs to be tightened here.
L16: why instead? again the logic is slippery; a variation with either problem solving OR personality traits seems to restrictive. The author might like to take a look at recent publications by Sol et al. 2016 and Griffin 2016, Phil Trans for a somewhat different view of this problem.
L21-24: To what part of Sih & Del Giudice 2012 is the author referring to? These authors never use the term behavioral flexibility in their paper so please explain in more detail how this paper can be used to make this assertion.
L24: As Sih & Del Giudice never refer to behavioral flexibility, this paper can only be cited here if behavioral flexibility is defined as ‘reversal learning’, which the Sih and Del Giudice do indeed relate to other traits. Please tighten the logic flow here.
L31: what is meant by individual behaviors? Personality traits? But reversal learning is then described. Please clarify.
L41: which predictions exactly?
L49: for clarity, please explain by what measure grackles were found to be behaviorally flexible in the previous study. Many animals are behaviorally flexible, so what does it mean to say that grackles are behaviorally flexible? Grackles are not particularly inventive relative to what? Unless reference is made to some data showing that grackles are more/less behaviorally flexible/innovative than other species and that the two traits are unrelated based on a reasonable sample size of grackles, I suggest cutting these comments as they seem somewhat arbitrary.
L264: common myna is just another name for Indian myna, so Sol et al 2012 and Griffin & Diquelou 2015 need to be collapsed.
L296: I am not sure what this last sentence means; please clarify. The birds only received a reward at the end of persisting so reward receipt per se cannot have influenced persistence.
L339: please refer to the exact dependent variable here rather than ‘behavioral flexibility’
L345-347: please cut this sentence in two to improve clarity
L373: but see Griffin et al. 2013 who showed a negative relation between reversal learning and problem solving.
L374-375: the lack of consistency might alternatively arise for the lack of consistency with which these traits are measured across the literature and the fact that repeatability of the traits is rarely established before cross-trait correlations are conducted.
L376-380: I am not sure what the author means to say with this last paragraph. What does distinct sources of variation mean? Is it meant that selection in an invasive species might have lead to a mechanistic dissociation between reversal learning and other traits, allowing reversal learning to evolve independently from other traits to support invasiveness? It would good to know exactly why grackles are considered invasive.

L25: flexible, not flexibile
L31: behaviours, not behavioura

---

## Round 0.2 · Minor Revisions

Thank you once again for being so responsive to the reviewers’ comments and for returning your revision promptly. I’ve had a careful read of your revision and I’m still finding some of the presentation of constructs somewhat confusing. In my comments below I am referring to page numbers from the clean proof rather than the tracked version.

I think the sentence on lines 12-14 is still unclear. Couldn’t behavioral flexibility be both a problem solving ability and arise because of links to other behaviors? I am not sure what you mean by this last clause in any case.

I still think that the opening paragraph could benefit from a cleaner discussion of when behavioral flexibility might lead to improvements in problem-solving and when it might not. Clarifying why the relationship might be inconsistent would be helpful for the reader and would hopefully place previous findings in context. Here is where the discussion on inhibition versus innovation would be useful. I’m not sure that the distinction between slow and fast learners makes sense to me. Fast learners might be more flexible if they are trying all kinds of various responses without the inhibition that delays novel responses in slower learners. I understand that your thinking along these lines is supported by previous theorizing in this area and that it informs your predictions but I wonder if you could provide a few clarifying sentences for other readers like me who might find the logic a bit counterintuitive. I think it makes more sense to distinguish whether behavioral flexibility is a more general category that subsumes other traits, or is a unique sub-component of something greater (“intelligence”? although this is not a word I like to use with non-human species comparisons). I say this because it isn’t clear to me what it means to say that behavioral flexibility performance could be caused by behavioral flexibility or by correlated behaviors (Lines 29-31). If behavioral flexibility subsumes a suite of other traits, that might all be correlated, then this distinction makes little sense.

On line 154, you write that you conducted a Spearman’s rank correlation test to determine if your two measures of exploration were distinct behaviors. I’m not sure this is how you want to state this. While showing a lack of perfect correlation might imply that the behaviors were not the same, the converse - a high correlation - does not imply that they are the same. You might argue for divergent validity instead. See also lines 270-271 and 303, 328-329, 379. For example, obesity may be highly correlated with depression but this does not mean these two variables are the same.

There should be a reference for the stick tool use experiment (lines 205 and 210).

On lines 85-86 you predict that birds that are more flexible should explore less and be less persistent but on lines 309-311 you write birds that were more flexible did not explore more and were not more persistent, which suggests the opposite prediction. In fact, then part of your findings are consistent with your predictions although you state them as if they were opposite.

On lines 328-329, shouldn’t there be a negative correlation between exploration and risk aversion? Are you treating non-aversive as risk aversion? That seems to be the case given your caption for Table 2 as well. This is confusing. Please change such that risk aversion is time spent away from the riskiest areas rather than time spent in the risky areas.

In future directions, you should consider a data reduction technique such as factor analysis or PCA to determine how different traits are related to each other, given a larger sample of individuals.

On line 78, change “while” to “whereas”

On line 395, insert a ; before “however”

Permissions might be needed to reproduce Figure 1, which is really a Table. I don’t think this Figure/Table is needed since the predictions are outlined in the text and the reader can be referred to the Sih and Del Giudice paper.

Remove the vertical lines from Table 3.

---

## Round 0.3 · accepted · Accept

Thank you once again for your prompt attention to the final requested revisions. This is a nice paper that will help clarify thinking about behavioral flexibility and how it might be measured.